

# Emotion brain network topology in healthy subjects following passive listening to different auditory stimuli

Muhammad Hakimi Mohd Rashid[1,2], Nur Syairah Ab Rani[2], Mohammed Kannan[2,3], Mohd Waqiyuddin Abdullah[2], Muhammad Amiri Ab Ghani[4], Nidal Kamel[5] and Muzaimi Mustapha[2]

[1] Department of Basic Medical Sciences, Kulliyyah of Pharmacy, International Islamic University, Kuantan, Pahang, Malaysia
[2] Department of Neurosciences, School of Medical Sciences, Universiti Sains Malaysia, Kubang Kerian, Kota Bharu, Kelantan, Malaysia
[3] Department of Anatomy, Faculty of Medicine, Al Neelain University, Khartoum, Khartoum, Sudan
[4] Jabatan Al-Quran & Hadis, Kolej Islam Antarabangsa Sultan Ismail Petra, Nilam Puri, Kota Bharu, Kelantan, Malaysia
[5] Centre for Intelligent Signal & Imaging Research (CISIR), Electrical & Electronic Engineering Department, Universiti Teknologi PETRONAS, Seri Iskandar, Perak, Malaysia

Corresponding authors
Muhammad Hakimi Mohd Rashid, hakimiuia@gmail.com
Muzaimi Mustapha, mmuzaimi@usm.my

## ABSTRACT

A large body of research establishes the efficacy of musical intervention in many aspects of physical, cognitive, communication, social, and emotional rehabilitation. However, the underlying neural mechanisms for musical therapy remain elusive. This study aimed to investigate the potential neural correlates of musical therapy, focusing on the changes in the topology of emotion brain network. To this end, a Bayesian statistical approach and a cross-over experimental design were employed together with two resting-state magnetoencephalography (MEG) as controls. MEG recordings of 30 healthy subjects were acquired while listening to five auditory stimuli in random order. Two resting-state MEG recordings of each subject were obtained, one prior to the first stimulus (pre) and one after the final stimulus (post). Time series at the level of brain regions were estimated using depth-weighted minimum norm estimation (wMNE) source reconstruction method and the functional connectivity between these regions were computed. The resultant connectivity matrices were used to derive two topological network measures: transitivity and global efficiency which are important in gauging the functional segregation and integration of brain network respectively. The differences in these measures between pre- and post-stimuli resting MEG were set as the equivalence regions. We found that the network measures under all auditory stimuli were equivalent to the resting state network measures in all frequency bands, indicating that the topology of the functional brain network associated with emotional regulation in healthy subjects remains unchanged following these auditory stimuli. This suggests that changes in the emotion network topology may not be the underlying neural mechanism of musical therapy. Nonetheless, further studies are required to explore the neural mechanisms of musical interventions especially in the populations with neuropsychiatric disorders.

## INTRODUCTION

Cognitive functions are the result of orchestrated symphony among networks of neuronal ensembles in cortical and subcortical structures (*Von Der Malsburg, 1994*). This insight has brought about a paradigm shift in neuroscience from focusing more on the reductionist approach of identifying an isolated unit responsible for a specific cognitive function to exploring and characterizing networks of brain functional units working together on performing cognitive tasks. Functionally cohesive networks formed by recruiting and coordinating spatially separated brain regions are central to the way in which brain processes and integrate information (*Schnitzler & Gross, 2005*).

Adopting connectivity approach, numerous research have been conducted to understand more on aspects of cognitive functions including but not limited to attention (*Fox et al., 2005*; *Fox et al., 2006*), memory (*Ranganath et al., 2005*) and emotional processing (*Kim et al., 2011*). Increasingly, the approach has also been incorporated in many studies to further elucidate diseases and pathologies in the brain functions. Brain connectivity researches enables researchers to gain deeper understanding and new insights on depression (*Connolly et al., 2013*; *Damborská et al., 2020*; *Benschop et al., 2022*), anxiety (*Kim et al., 2011*; *Al-Ezzi et al., 2020*; *Betrouni et al., 2022*), schizophrenia (*Lynall et al., 2010*; *Alamian et al., 2020*; *Mackintosh et al., 2021*), epilepsies (*Van Mierlo et al., 2014*; *Dharan et al., 2021*; *Routley et al., 2020*) and many more. Connectivity studies have demonstrated that these pathologies are the result of abnormalities of large scale brain networks rather than dysfunction of a specific brain region. Furthermore, this has also opened up ways of exploring new approaches to treatment and ways to evaluate effectiveness of treatments (*Huang et al., 2023*; *Aydin et al., 2020*).

In depression and anxiety specifically, music and other rhythmic auditory stimuli are among the non-pharmacological approach that have been used either alone or in combination with pharmacological interventions and psychotherapy. A meta-analysis of nine clinical trials involving 421 participants found moderate-quality evidence that musical therapy improve depressive symptoms, reduce anxiety levels and improve functioning in individuals with depression (*Aalbers et al., 2017*). Another meta-analysis, incorporating 55 randomized control trials, demonstrated that both music therapy and music medicine significantly reduce depressive symptoms, with music medicine demonstrating a more pronounced effect (*Tang et al., 2020*). Music therapy has also been shown to improve anxiety in critical care settings (*Erbay Dalli, Bozkurt & Yildirim, 2023*; *Bro et al., 2018*; *Chen et al., 2023*).

Music perception is known to activate multiple cortical areas (*Koelsch, 2011*). It has been shown to simultaneously activate auditory and reward systems in the brain (*Quinci et al., 2022*). Several studies have also looked into the information flow and coordination among these and other cortical regions (*Zhu et al., 2023*; *Carriere et al., 2020*; *Wu et al., 2019*; *Karmonik et al., 2016*). It has been shown that clustering coefficient was larger and characteristic path length was smaller while listening to musical stimuli (*Qiu et al., 2022*; *Wu et al., 2012*). These network metrics characterize functional segregation and functional integration of the network respectively. Networks with relatively large clustering coefficient

**Table 1  List of auditory stimuli and their rhythmic characteristics.**

| Stimuli | | |
|---|---|---|
| 1 | Monochord sounds | Rhythmic |
| 2 | Hare Krishna | Rhythmic |
| 3 | Arabic News | Non-rhythmic |
| 4 | Arabic Poem | Rhythmic |
| 5 | Al-Kursi | Rhythmic |

and small characteristic path length are called small world networks. These networks are efficient at both local information processing and global information transfer (*Latora & Marchiori, 2001*). On the contrary, brain functional networks tend to diverge from small world network in depression (*Teng et al., 2022*; *Zhang et al., 2020*; *Guo et al., 2014*; *Li et al., 2015*) and other brain pathology (*Qi et al., 2023*; *Sanz-Arigita et al., 2010*; *Shim et al., 2014*). However, the underlying neural mechanisms for musical therapy remain elusive (*Maratos, Crawford & Procter, 2011*; *Chan & Han, 2022*). In particular, the effect of musical therapy on brain networks that are involved in emotion processing have not been extensively studied. Furthermore, most studies on music therapy has predominantly focused on Western music, leaving a noticeable gap in the exploration of other forms of rhythmic auditory stimuli.

In this study, we explored five different auditory stimuli (Table 1) and their effects on the functional brain network that are associated with emotion processing as the the potential neural underpinning of the therapeutic effects of musical intervention using a MEG recording. To our best knowledge, no such study had been conducted, either in healthy individuals or in those with neuropsychiatric pathologies. We conducted the study among healthy subjects since it is necessary to establish a referential range among healthy individuals from which further study may refer to.

The aim of this study, therefore, is to ascertain the presence or absence of the effects of different auditory stimuli on the topology of functional brain networks that are linked to emotional regulation and dysregulation using graph theoretic measures. Specifically, the effects of these stimuli on the mentioned brain network in terms of its local and global information processing efficacy as measured by transitivity and global efficiency. Thus, the null hypothesis of this study posits that there are no changes in the topological measures weighted transitivity T and global efficiency E of the brain network involved in emotion processing across all different auditory stimuli for all frequency bands: delta, theta, alpha, beta, and gamma. On the other hand, the alternate hypothesis suggests that specific auditory stimuli may indeed lead to significant alterations in these brain network measures. By looking at the changes in these network metrics, we hoped to gain a better understanding on how these auditory stimuli may impact brain emotion processing function and potentially inform the underlying neural mechanism of the therapeutic interventions. We approached this using naturalistic paradigm where subjects' MEG were recorded while they were passively listening to the auditory stimuli. This reflects more

closely to the real-world scenario of how music is being consumed and how it is applied in therapeutic settings.

## METHODS

### Study designs

The study was conducted using a cross over experimental design in which all subjects were exposed to a series of five auditory stimuli in specified orders. The order of sequences in which auditory stimuli were presented was generated in such a way that each stimuli must occur once within each subject, and each stimuli must have the same number of occurrence. This was done to control the period and order effects in the experiment. This study was conducted in accordance with the Helsinki declaration. It was reviewed and approved by the institutional Human Research Ethics Committee of Universiti Sains Malaysia (FWA Reg No: 00007718; IRB Reg No: 00004494). In addition, we followed the recommendations laid out by the guidelines for conducting and reporting MEG research (*Gross et al., 2013*; *Keil et al., 2014*). The sample size for this study was estimated using G* power and satisfied statistical power of 95% at a 0.05 two-sided significance level. The sample size was estimated based on the data observed in a previous study (*Wu et al., 2012*). The estimated total sample size for this study was 26. To mitigate potential attrition, we eventually decided to include 30 subjects in the study.

In this study, 32 subjects underwent screening. However, two individuals were excluded due to psychiatric illnesses, specifically, schizophrenia and generalized anxiety disorder. Thus, our final study cohort comprised 30 enrolled subjects (aged 21–35 years old). 15 of them were male and 15 were female. All of them were non-Arabic speakers. They have no known neurological disease, history of head trauma, hearing problem or psychiatric illnesses. None have history of illicit drugs abuse. All subjects provided written informed consent prior to the study. Each subject was advised to have an adequate sleep of 8 h every night for 1 week prior to the study. They were also advised not to consume any caffeinated foods and beverages, alcohols, and other psychoactive substances. Subjects with any metal containing implants were excluded from this study. Flow chart in Fig. 1 summarise the whole process of subject recruitment and MEG acquisition.

### Stimuli

Five different auditory stimuli were considered in this study. The session starts with 3 min recording of resting state where the subjects were asked to sit comfortably with their eyes closed without being presented with any external stimuli or engaging in any activity. A specified order of five different auditory stimuli were subsequently administered for the total length of 3 min per stimuli. The stimuli were administered with Presentation software (Neurobehavioral Systems, Ltd., Berkeley, CA, USA). The sound was delivered through a pair of pneumatic headphones at individually adjusted loudness. In between stimuli, there was a 1-minute rest period where no auditory stimuli were given. The session ended with 3-minutes of resting state recording.

Both rhythmic and non-rhythmic auditory stimuli were considered in this study. Table 1 lists all the stimuli used in this study and their characteristics. Each stimulus contributes

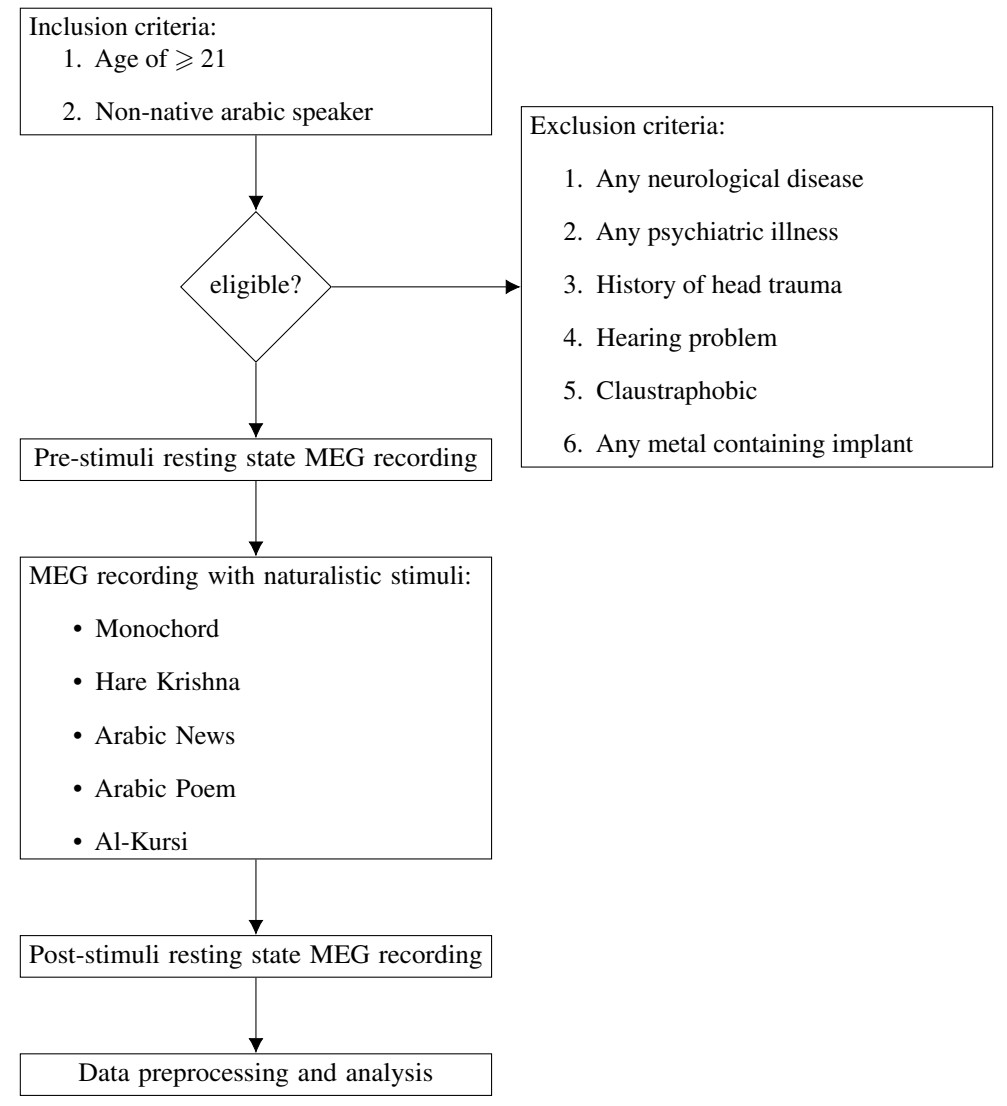

Figure 1 Flow chart of subject recruitment and MEG acquisition.

to the overall understanding of the neural mechanisms of music therapy. Monochord is an instrument having all of its 30 strings tuned to one base tone, producing varying overtones sounds that merge into one continuous sound without any specific scale and harmony. The produced sounds have been shown to have relaxation effects in children with anxiety (*Goldbeck & Ellerkamp, 2012*). Similar findings was also noted in patients who underwent chemotherapy (*Lee et al., 2012*). The monochord's single-tone sound could help in understanding how simple, repetitive sounds affect the brain's emotion network. Hare Krishna is one of the mantra recited in Hindu religion as part of meditation practice. Meditation has been shown to be an effective mean of emotional control (*Braboszcz, Hahusseau & Delorme, 2010*). Its inclusion helps in understanding how a melodious religious chant affects the neural network regulating emotion. Al-Kursi is the 225th verse of

the 2nd chapter in the Quran. The verse was recited by a renown reciter and was recorded in a dedicated sound-proof Audio Room, School of Medical Sciences, USM Health Campus. This verse is used in Ruqyah which is a method of treatment using Quranic verses and supplications practised within the the muslim community (*Haque & Keshavarzi, 2014*; *Abu-Rabia, 2005*). Its inclusion in the study can shed light on how melodious recitation might influence emotion processing brain network informing the neural mechanism of its therapeutic effects.

Apart from the obvious reason of being composed in the same language, Arabic news and poems were included as the control of the Al-Kursi stimuli since their sound originated from the same instrument as the Quranic recitations, *i.e.,* human vocal cord. Arabic poem titled "If my Lord asks me" was chosen for its rhythmic features which is comparable to the rhythmic features of the Quranic recitation, while Arabic news was chosen since in contrast to Quranic recitation, it is non-rhythmic in nature. By analyzing the different responses these stimuli evoke in the brain's emotion network, the study aims to explore the neural underpinnings of music therapy. This could potentially lead to more targeted and personalized therapeutic interventions. The audio files of all stimuli used in the present study were available online (see Data Availability section).

## Acquisition and preprocessing of MEG datasets

Simultaneous EEG and MEG data were recorded at the MEG laboratory of Universiti Sains Malaysia. The recording were conducted in an electrically and magnetically shielded room with Vectorview™ 306-channel MEG system (Elekta Neuromag; Elekta Oy, Helsinki, Finland) combined with a compatible 64-channels EEG system. The MEG system consist of 204 gradiometers and 102 magnetometers. Subjects were instructed to remove any metals, electronics or metal containing clothing before entering the room. The recording sample rate was set at 1000 Hz. The reference and ground electrode were put on the nose tip and on the right cheek respectively. Eye movements and blinks, were recorded with 2 electrooculogram (EOG) electrodes attached close to the external eye corners on both sides. Electrocardiogram (ECG) was also simultaneously recorded for the purpose of removing artifacts of cardiac origin. Four head position indicator (HPI) coils were embedded within the head cap and used to record head movement and subsequently used for movement artifact correction. These locations were determined relative to the nasion, two preauricular points and more than 100 additional points on the surface of scalp and nose which were recorded by the Polhemus Isotrak 3D digitizer. Through out experiment, subjects were comfortably seated with eyes closed. Overall, we have the recordings of 30 subjects for each stimuli except the 8th stimuli and post resting state where there were recordings from only 27 subjects.

MEG data was preprocessed using Brainstorm software (*Tadel et al., 2011*) and MNE software (*Gramfort et al., 2014*). A series of preprocessing steps were done in order to suppress or remove noise and artifacts. Bad channels were identified and excluded from analysis. We then applied notch filter to remove power line noise at 50 Hz, 100 Hz, and 150 Hz peaks. The temporal signal space separation (tSSS) were used to suppress noise originated from outside of the MEG dewar and to correct for head motion artifacts (*Taulu*

*& Simola, 2006*). The following setting parameters were used: length of correlation window, 30 s; subspace correlation limit, 0.98; order of internal component of spherical expansion, 8; and order of external component of spherical expansion, 3. Physiological artifacts of cardiac origin, eye movements, eye blinks and muscular activity were isolated and removed using independent component analysis (ICA) method (*Barbati et al., 2004*). The resulting MEG were manually inspected to ensure satisfactory artifacts correction and also to identify any remaining or additional bad segments which were then excluded from subsequent analysis.

## Source localization

Source localization was done using Brainstorm software (*Tadel et al., 2011*) by following the pipeline described in *Niso et al. (2019)*. Specifically, we used fiducial and digitization points of each subjects to warp the template MRI of International Consortium for Brain Mapping (ICBM152) (*Fonov et al., 2009*) in order to create a psudo-individual anatomy. The warped MRIs were then coregistered with the MEG sensors for each recording sessions. We check the result to ensure satisfactory alignment and orientation of the MEG sensors and the warped MRI.

We modelled the remaining sensors and environmental noise as noise covariance matrix using 2 min of empty room MEG recordings. We then defined forward model for each recording sessions using the overlapping spheres method (*Huang, Mosher & Leahy, 1999*). The forward or head models describe how neuronal activity at the cortical sources is being transformed to the magnetic flux measured at the level of MEG sensors. In order to capture neuronal activity at cortical surfaces as well as at deep subcortical structures, we defined source space as a set of distributed dipoles of the whole brain volume with unconstrained dipoles orientation.

Once forward or head models were defined for each recording sessions, source estimation was carried out for each of them using depth weigthed minimum norm estimation (wMNE) (*Lin et al., 2006*). This method has been validated in many studies using either simulation or by comparing the estimation with intracranial EEG recording (*Halder et al., 2018*; *Afnan et al., 2023*; *Mikulan et al., 2020*; *Pascarella et al., 2023*). The depth weighting parameter was set at 0.75 as suggested by *Lin et al. (2006)* in order to minimize the localization error particularly for the deeper subcortical structures. The resulting inverse operators were then used to reconstruct time series at the level of brain regions of interest. We used automated anatomical labelling atlas 3 (AAL3) (*Rolls et al., 2020*) to segment volume source space and define brain regions of interest. The time series for each brain regions were computed as the mean of all voxels within each regions.

## Brain regions of interest

We considered 61 brain regions that have been shown in the literature to be involved in emotional regulation (*Banks et al., 2007*; *Cisler et al., 2013*; *Kohn et al., 2014*). Among these are brain regions that were stipulated to be involved in experiencing musical emotions (*Alluri et al., 2015*). Dysfunction of these regions have also been shown to be associated with several neuropsychiatric pathologies including depression (*Greicius et al., 2007*; *Veer*

**Table 2  List of all brain regions of interest.**

| Brain regions | | |
|---|---|---|
| L Amygdala | L Ant OFC | L VTA |
| R Amygdala | L Lat OFC | R VTA |
| L Angular | R Lat OFC | Vermis |
| R Angular | L Med OFC | Thalamus |
| L Calcarine S | R Med OFC | Nucleus Accumbens |
| R Calcarine S | L Post OFC | L Lingual |
| L Caudate Nucleus | R post OFC | R Lingual |
| R Caudate | L ParaHippocampal C | L ACC |
| L MCC | R ParaHippocampal C | R ACC |
| R MCC | L Inf Parietal C | L Cerebellum |
| L PCC | R Inf Parietal C | R Cerebellum |
| R PCC | L Sup Parietal C | L Cerebellum Crus |
| L Mid Frontal C | R Sup Parietal C | R Cerebellum Crus |
| R Mid Frontal C | L Putamen | L Inf Frontal C |
| L Sup Frontal C | R Putamen | R Inf Frontal C |
| R Sup Frontal C | L Inf Temporal C | L Vm Frontal C |
| R Hippocampus | R Inf Temporal C | R Vm Frontal C |
| R Hippocampus | L Mid Temporal | L Dm Frontal C |
| L Insula | R Mid Temporal c | R Dm Frontal C |
| R Insula | L Sup Temporal C | R Sup Temporal C |
| | L Ant OFC | |

**Notes.**

L, Left; R, Right; C, Cortex; Ant, Anterior; Mid, Middle; Med, Medial; Inf, Inferior; Post, Posterior; Sup, Superior; OFC, OrbitoFrontal Cortex; ACC, Anterior Cingulate Cortex; MCC, Middle Cingulate Cortex; PCC, Posterior Cingulate Cortex; Vm, Ventromedial; Dm, Dorsomedial.

*et al., 2010*; *Lui et al., 2011*), obsessive compulsive disorder (*Harrison et al., 2009*; *Sakai et al., 2011*; *Hou et al., 2014*; *Takagi et al., 2017*), and anxiety (*Etkin et al., 2009*; *Kim et al., 2011*; *Hahn et al., 2011*). We have also included brain regions that have been used in functional neuroimaging studies to monitor and assess treatment responses in these disorders (*Crowther et al., 2015*; *Walsh et al., 2017*; *Scult et al., 2019*; *Walsh et al., 2019*). Table 2 lists all the brain regions that were included in the analysis. These brain regions were specified by selecting and merging together brain regions defined in AAL3.

## Connectivity analysis

For the purpose of our study, we applied amplitude envelope correlation (AEC) to extract bivariate connectivity strength between the pairs of all to all $61 \times 61$ brain regions (*Bruns et al., 2000a*; *Bruns et al., 2000b*). Connectivity strength were computed at the level of brain regions using time series estimated for each brain regions as described in source localization section.

The time series were first band pass filtered to frequency bands of interest: delta ($\delta$) ($2-4\ Hz$), theta ($\theta$) ($5-7\ Hz$), alpha ($\alpha$) ($8-12\ Hz$), beta ($\beta$) ($15-29\ Hz$), and gamma ($\gamma$) ($30-59\ Hz$). This frequency ranges were the convention used in Brainstorm software and have been used in many prior studies (*Gehrig et al., 2012*; *Wiesman et al., 2022*;

*Rempe et al., 2023*; *Bergwell et al., 2023*; *Nugent et al., 2020*). In order to avoid spurious connectivity between the reconstructed source time series caused by signal leakage, we applied orthogonalized AEC where time series were orthogonalized prior to computation of AEC (*Hipp et al., 2012*). Orthogonalization of time series was performed in frequency domain before computing their power envelopes. A complex signal $y_{\perp x}(t,f)$ is orthogonal to signal $x(t,f)$ as defined by the following equation:

$$y_{\perp x}(t,f) = \Im\left(\frac{x(t,f)^*}{|x(t,f)|}y(t,f)\right) \tag{1}$$

where $\Im$ denotes imaginary part of the complex signals and $x(t,f)^*$ is the complex conjugate of $x(t,f)$.

Analytical signal $z(t)$ were then constructed using Hilbert transform,

$$z(t) = x(t) + iH[x(t)] \tag{2}$$

$$H[x(t)] = \frac{1}{\pi}PV\int_x^\infty \frac{x(\tau)}{t-\tau}d\tau \tag{3}$$

where $i$ is the imaginary part, $H[x(t)]$ is the Hilbert transform and $PV$ is the Cauchy principle value. The amplitude envelope were then computed as follows:

$$a(t) = \sqrt{(x(t))^2 + (H[x(t)])^2} \tag{4}$$

With amplitude envelope of source time series obtained, connectivity (AEC) between amplitude envelope of $i$th brain region $\mathbf{a_i}$ and $j$th brain region $\mathbf{a_j}$ was computed as a Pearson correlation coefficient $\rho$ between them:

$$\rho(\mathbf{a_i}, \mathbf{a_j}) = \frac{\mathbf{a_i}\mathbf{a_j}^{\mathbf{T}}}{\sqrt{\mathbf{a_i}\mathbf{a_i}^{\mathbf{T}}}\sqrt{\mathbf{a_j}\mathbf{a_j}^{\mathbf{T}}}} \tag{5}$$

where $\mathbf{a_i}$ and $\mathbf{a_j}$ are $1 \times T$ vectors of mean centered amplitude envelope of $i$th and $j$th brain regions respectively. $\mathbf{a_i}^{\mathbf{T}}$ and $\mathbf{a_j}^{\mathbf{T}}$ are both transposes of $\mathbf{a_i}$ and $\mathbf{a_j}$. Correlation coefficient of all to all brain regions were organized into matrix forming connectivity or weighted adjacency matrix $\mathbf{A}$.

## Network analysis

Once connectivity matrices $\mathbf{A} \in \mathbb{R}^{61 \times 61}$ with elements $w_{ij}$ representing connectivity strength (AEC) between $i$th and $j$th brain regions were computed for each subjects, stimuli and frequency, we summarized the results using tools developed in graph theory. Graph theory offers several tools to characterize the topological features of complex brain networks. In order to ensure robust findings, the connectivity matrices were thresholded leaving out small and insignificant connectivity values which can be considered as noises.

We adapted the recommendations proposed by *Cohen (2014)* to threshold connectivity matrices using absolute thresholding method with median value of connectivity strength as the cut-off point. In order to ensure the threshold is independent of each stimuli, we

determined the threshold value based on pooled values of connectivity strength from all stimuli. We determined specific threshold separately for each frequency bands since connectivity values differ across frequency bands.

In this study, we computed two topological features from the thresholded connectivity matrices using the Brain Connectivity Toolbox (BCT) (*Rubinov & Sporns, 2010*). These are weighted transitivity and weighted global efficiency. Weighted transitivity and global efficiency are particularly useful and relevant measures in this study since they allow for the assessment of the efficiency of both local information processing and global information transfer respectively.

Weighted transitivity is a variant of mean clustering coefficient which reflects the average prevalence of clustered connectivity around individual nodes. Weighted transitivity $T$ is measured using the following formula:

$$T = \frac{\sum_i^N 2t_i}{\sum_i^N k_i(k_i - 1)} \tag{6}$$

where $k_i$ is the number of links connected to node $i$. $k_i(k_i - 1)$ is thus the total number of possible links that could exist among the vertices. $N$ is the total number of nodes in the network and $t_i$ is the weighted geometric mean of triangle around node $i$ computed as follows:

$$t_i = \frac{1}{2} \sum_{j,h}^N (w_{ij} w_{ih} w_{jh})^{\frac{1}{3}} \tag{7}$$

where $w_{ij}$ is the connectivity strength from node $i$ to node $j$. All self connection strength $w_{ii}$ were set to zero ($w_{ii} = 0$ for all $i$).

Weighted global efficiency $E$ measures the efficacy of the brain network in integrating information from distributed brain regions and is calculated using the following formula:

$$E = \frac{1}{N} \sum_i^N \frac{\sum_{j,j \neq i}^N (d_{ij})^{-1}}{N - 1} \tag{8}$$

where $d_{ij}$ is the shortest path length between nodes $i$ and $j$ calculated as follows:

$$d_{ij} = \sum_{w_{uv} \in g_{i \rightarrow j}} f(w_{uv}) \tag{9}$$

where $f$ is a map from connectivity weight to length and $g_{i \rightarrow j}$ is the shortest weighted path between node $i$ and node $j$.

## Statistical analysis

The analysis were done using R statistical programming language (*R Core Team, 2020*). We analysed the data using Bayesian linear mixed model. Brms package were used to fit the model (*Bürkner, 2017*). Brms is the R wrapper for Stan which is another probabilistic programming language that implement Hamiltonian Monte Carlo (HMC) sampling algorithm (*Carpenter et al., 2017*). Following the recommendation by *Barr et al. (2013)*, the model include the maximal random effects structure justified by the experimental design,

*i.e.,* varying intercept and slope model with subjects as the grouping variable. Outliers were identified using Boxplot method where all data points that fall outside the threshold of 1.5 times the interquartile range below the first quartile or above the third quartile and were excluded before model fitting. As part of sensitivity analysis, we have also performed the statistical analysis while including the outliers. We have also conducted sensitivity analysis on strategies for handling missing data by using both only complete cases and multiple imputations to account for missing data. In line with our research objectives and experimental design, we fitted the following model for each network measures:

$$y_{i|s} \sim Log\mathcal{N}(\mu_s, \sigma) \tag{10}$$

$$\mu_s = \alpha_s + \sum_j \beta_{s,j} Stimuli_j \tag{11}$$

$$\begin{bmatrix} \alpha_s \\ \beta_{s,j} \end{bmatrix} \sim \text{MV Normal}\left( \begin{bmatrix} \bar{\alpha} \\ \bar{\beta} \end{bmatrix}, \mathbf{Q} \right) \tag{12}$$

$$\mathbf{Q} = \begin{bmatrix} \sigma_\alpha & 0 \\ 0 & \sigma_\beta \end{bmatrix} \mathbf{R} \begin{bmatrix} \sigma_\alpha & 0 \\ 0 & \sigma_\beta \end{bmatrix} \tag{13}$$

$$\bar{\alpha} \sim \mathcal{N}(\bar{\mu}_k, 0.2) \tag{14}$$

$$\bar{\beta} \sim \mathcal{N}(0, 0.2) \tag{15}$$

$$\sigma_\alpha \sim half\,\mathcal{N}(0, 0.1) \tag{16}$$

$$\sigma_\beta \sim half\,\mathcal{N}(0, 0.1) \tag{17}$$

$$\mathbf{R} \sim LKJcorr(2) \tag{18}$$

$$\sigma \sim half\,\mathcal{N}(0, 0.1) \tag{19}$$

where $y_{i|s}$ refers to $i$th observation of dependent variables of interest (network measures - $T$ and $E$) of subject $s$. $Log\mathcal{N}(\mu, \sigma)$ is the notation for log-normal distribution with location

μand scale $\sigma$. $\alpha_s$ is the specific intercept for subject $s$ (pre-stimuli resting state). $\beta_{s,j}$ is the coefficient parameter for $j$th stimulus (how much the stimulus differs from the pre-stimuli resting state recording; here the stimuli include post-stimuli resting state recordings) in subject $s$. Both $\alpha_s$ and $\beta_{s,j}$ are jointly distributed with multivariate normal distribution around the overall intercept $\bar{\alpha}$ and slope $\bar{\beta}$. $\mathbf{Q} \in \mathbb{R}^{2 \times 2}$ is the covariance matrix of the multivariate Gaussian distribution defined as in (Eq.13). $\sigma_\alpha$ and $\sigma_\beta$ are standard deviations of $\alpha_s$ and $\beta_{s,j}$ respectively. $\mathbf{R} \in \mathbb{R}^{2 \times 2}$ is the correlation matrix. Both Eqs. (10) and (11) define the likelihood function and the rest (Eqs.12–19) define priors of our generative model. Additionally, as a component of our sensitivity analysis, we have also utilized the Student's t-distribution to model the data. We have also tested the sensitivity of using non-informative priors where $\bar{\alpha} \sim \mathcal{N}(\bar{\mu}_k, 10), \bar{\beta} \sim \mathcal{N}(0, 100), \sigma_\alpha \sim half \mathcal{N}(0, 10), \sigma_\beta \sim half \mathcal{N}(0, 10),$ and $\sigma \sim half \mathcal{N}(0, 10)$.

We set weakly informative priors (*McElreath, 2020*) on the parameters $\bar{\beta}$ i.e the overall slope of the stimuli, the standard deviation of the intercept $\sigma_\alpha$, the standard deviation of the slope $\sigma_\beta$, the correlation matrix $\mathbf{R}$ and the standard deviation of the models $\sigma$. For the purpose of our research, we set the prior for $\bar{\beta}$ so that the slopes values are symmetric around 0 with finite variance. We set priors for parameter $\bar{\alpha}$ to be normally distributed around each network measure mean $\bar{\mu}_k$ with finite variance. We checked the prior predictive distributions to ensure the chosen priors make sense and generate plausible and realistic data prior to observations.

Four sampling chains were used to explore and sample from the posterior distributions for each model. These chains ran for 14,000 iterations with 4,000 warm-up iterations yielding 40,000 samples for each parameters. Several HMC diagnostics were monitored to determine the convergence of numerical integration of posterior distribution. These include the potential scale reduction factor $\hat{R}$, traceplots and effective sample size (ESS). $\hat{R}$ of $< 1.1$ indicates convergence of the sampling procedure and EES should be $> 10,000$ samples for stable posterior estimates (*Kruschke, 2014*). Once we fit the model, we checked the posterior predictive distributions to ensure that the model reflects the observed data.

We set equivalence regions based on principles that were laid out in *Kruschke (2018)*; *Kruschke & Liddell (2018)*; *Kruschke (2011)*. We proceed by specifying the ROPE which is the range of values within which the difference between two conditions or stimuli is taken to be small enough to be practically equivalent. Specifically, we used the total credible interval (CI) of the difference between pre-stimuli resting state and post-stimuli resting state recordings to specify the ROPE. This ensures that the equivalence region covers the whole interval of the differences between the two resting states. The 90% CIs for each stimulus and their differences were defined using the 90% highest density intervals (HDI). These HDI were then used together with ROPE to test for the presence or absence of effects. The network measures for each stimulus were considered equivalence to the resting state recording (absence of effects) if the entire 90% CI of the difference (*i.e.,* stimuli—pre-resting) lie within the equivalence region. If the entire 90% CI of the difference was outside the ROPE, the null hypothesis of equivalence or absence of effects should be rejected in favour of the alternate hypothesis of difference or presence of effects. If the 90% CI was neither completely lie within nor completely lie outside the ROPE, the

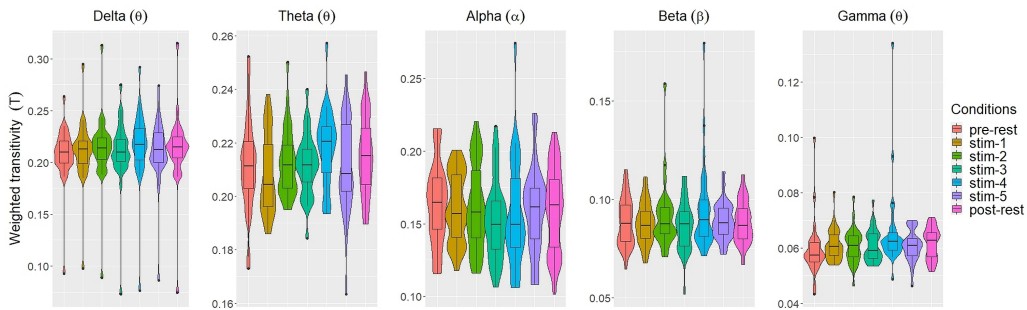

**Figure 2** The violin plots and boxplots illustrate the distribution of weighted transitivity *T* for each stimulus across different frequency band.

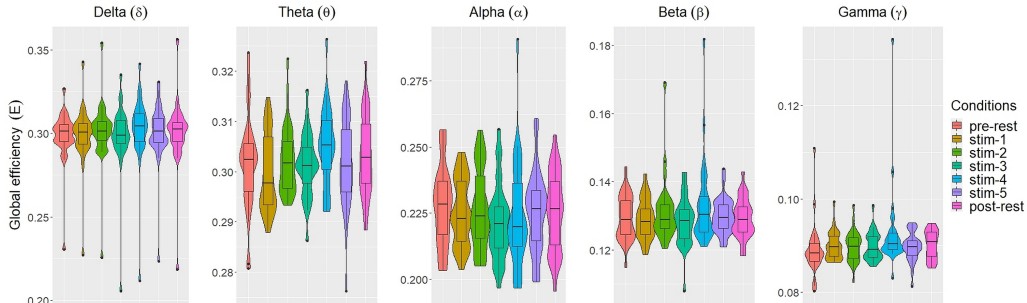

**Figure 3** The violin plots and boxplots illustrate the distribution of global efficiency *E* for each stimulus across different frequency band.

presence or absence of effects was undecidable. We reported the means and 90% CI under posterior distributions for the differences between each parameters.

## RESULTS

Figs. 2 and 3 depicted violin plots together with the embedded boxplots of weighted transitivity *T* and global efficiency *E* respectively under all experimental conditions for each frequency bands: $\delta, \theta, \alpha, \beta$ and $\gamma$. The boxplots displayed the following data summary: minimum, 1st quartile, median, 3rd quartile and maximum. Black dots below and above the minimum and maximum values were the outliers. Violin plots represent the distribution of values. Within each frequency bands, the distribution of the weighted transitivity *T* (Fig. 2) for each conditions were within intervals that were quite similar to each other with the exception of a few outliers.

Similar trend were also seen for global efficiency *E* (Fig. 3). Within each frequency bands, the values of *E* for each experimental conditions were similar to each other. A few outliers were also present. These were excluded in subsequent analysis.

The values of each network measures for each experimental conditions varied across frequency bands. Within each experimental conditions, the values of all network measures progressively decreased as the frequency band increased from $\delta$ to $\gamma$.

**Table 3  Equivalence test of transitivity T for each stimuli relative to pre-resting state across all frequency bands.**

| Frequency bands | Contrast | Median | 90% CI | %inside ROPE | Equivalence | $\hat{R}$ | ESS |
|---|---|---|---|---|---|---|---|
| Delta ROPE:<br>[−0.0127–0.0173] | Stim01 - Pre | 0.0013 | [−0.0048–0.0072] | 100% | Accepted | 1.00 | 22043 |
|  | Stim02 - Pre | 0.0039 | [−0.0026–0.0102] | 100% | Accepted | 1.00 | 22789 |
|  | Stim03 - Pre | 0.0026 | [−0.0037–0.0088] | 100% | Accepted | 1.00 | 21475 |
|  | Stim04 - Pre | 0.0084 | [0.0021–0.0149] | 100% | Accepted | 1.00 | 21349 |
|  | Stim05 - Pre | 0.0039 | [0.0026–0.0104] | 100% | Accepted | 1.00 | 21457 |
| Theta ROPE:<br>[−0.0108–0.0183] | Stim01 - Pre | −0.0027 | [−0.0083–0.0031] | 100% | Accepted | 1.00 | 22177 |
|  | Stim02 - Pre | 0.0016 | [−0.0039–0.0067] | 100% | Accepted | 1.00 | 21578 |
|  | Stim03 - Pre | 0.0006 | [−0.0048–0.0059] | 100% | Accepted | 1.00 | 21582 |
|  | Stim04 - Pre | 0.0059 | [0.0004–0.0114] | 100% | Accepted | 1.00 | 21575 |
|  | Stim05 - Pre | 0.0016 | [−0.0044–0.0071] | 100% | Accepted | 1.00 | 21763 |
| Alpha ROPE:<br>[−0.0210 0.00691] | Stim01 - Pre | −0.0016 | [−0.0087–0.0049] | 100% | Accepted | 1.00 | 30720 |
|  | Stim02 - Pre | −0.0020 | [−0.0106–0.0067] | 100% | Accepted | 1.00 | 26398 |
|  | Stim03 - Pre | −0.0102 | [−0.0168, −0.0033] | 100% | Accepted | 1.00 | 30368 |
|  | Stim04 - Pre | −0.0114 | [−0.0180, −0.0043] | 100% | Accepted | 1.00 | 30221 |
|  | Stim05 - Pre | −0.0039 | [−0.0119–0.0038] | 100% | Accepted | 1.00 | 28364 |
| Beta ROPE:<br>[−0.01–0.01] | Stim01 - Pre | −0.0003 | [−0.0036–0.0033] | 100% | Accepted | 1.00 | 17773 |
|  | Stim02 - Pre | 0.0008 | [−0.0017–0.0035] | 100% | Accepted | 1.00 | 24387 |
|  | Stim03 - Pre | −0.0004 | [−0.0030–0.0023] | 100% | Accepted | 1.00 | 24423 |
|  | Stim04 - Pre | 0.0009 | [−0.0016–0.0032] | 100% | Accepted | 1.00 | 25290 |
|  | Stim05 - Pre | 0.0008 | [−0.0017–0.0032] | 100% | Accepted | 1.00 | 25925 |
| Gamma ROPE:<br>[0.00–0.01] | Stim01 - Pre | 0.0026 | [0.0010–0.0043] | 100% | Accepted | 1.00 | 24198 |
|  | Stim02 - Pre | 0.0020 | [0.0002–0.0040] | 100% | Accepted | 1.00 | 23328 |
|  | Stim03 - Pre | 0.0024 | [0.0006–0.0041] | 100% | Accepted | 1.00 | 24653 |
|  | Stim04 - Pre | 0.0039 | [0.0023–0.0055] | 100% | Accepted | 1.00 | 24003 |
|  | Stim05 - Pre | 0.0026 | [0.0010–0.0044] | 100% | Accepted | 1.00 | 24658 |

In order to test for the absence (equivalence) or presence of effects, we ran the statistical analysis as discussed in the previous section. The $\hat{R}$ values which indicate convergence of the MCMC chains for all parameters for all models were less than 1.1 (Tables 3 and 4). As can be seen from these tables, the effective length of MCMC chains as indicated by EES for all parameters were more than 10,000 samples which provided reasonably stable estimates of the CI limits. From the density plot of observed and simulated data, it can be seen that the chosen prior for each model generated simulated data that approximate observations where most values were concentrated around the observed data with small probability of producing extreme values (refer to Fig. S1 and Fig. S3 in the supplementary materials). Posterior predictive check revealed that each fitted model provided a relatively accurate description of the observed data (refer to Fig. S2 and Fig. S4 in the supplementary materials).

Figure 4 showed the distribution of weighted transitivity $T$ differences between each auditory stimuli and pre-resting state. The shaded area between dash lines indicated the ROPE defined as stated in statistical analysis section. The red shaded area of each

**Table 4  Equivalencee test of global efficiency *E* for each stimuli relative to pre-resting state across all frequency bands.**

| Frequency Bands | Contrast | Median | 90% CI | %inside ROPE | Equivalence | $\hat{R}$ | ESS |
|---|---|---|---|---|---|---|---|
| Delta ROPE: [−0.0080–0.0104] | Stim01 - Pre | 0.0006 | [−0.0040–0.0028] | 100% | Accepted | 1.00 | 21174 |
| | Stim02 - Pre | 0.0009 | [−0.0027–0.0047] | 100% | Accepted | 1.00 | 19763 |
| | Stim03 - Pre | 0 | [−0.0035–0.0036] | 100% | Accepted | 1.00 | 21364 |
| | Stim04 - Pre | 0.0038 | [0.0002–0.0037] | 100% | Accepted | 1.00 | 20965 |
| | Stim05 - Pre | 0.0014 | [−0.0022–0.0051] | 100% | Accepted | 1.00 | 21659 |
| Theta ROPE: [−0.0066–0.0100] | Stim01 - Pre | −0.0021 | [−0.0052–0.0007] | 100% | Accepted | 1.00 | 19970 |
| | Stim02 - Pre | 0.0002 | [−0.0025–0.0032] | 100% | Accepted | 1.00 | 20325 |
| | Stim03 - Pre | −0.0002 | [0.0030–0.0026] | 100% | Accepted | 1.00 | 20562 |
| | Stim04 - Pre | 0.0026 | [−0.0023–0.0055] | 100% | Accepted | 1.00 | 20502 |
| | Stim05 - Pre | 0.0008 | [0.0021–0.0040] | 100% | Accepted | 1.00 | 20584 |
| Alpha ROPE: [−0.0129–0.0075] | Stim01 - Pre | −0.0019 | [−0.0052–0.0015] | 100% | Accepted | 1.00 | 24958 |
| | Stim02 - Pre | −0.0010 | [−0.0055–0.0039] | 100% | Accepted | 1.00 | 19458 |
| | Stim03 - Pre | −0.0058 | [−0.0093–0.0024] | 100% | Accepted | 1.00 | 24291 |
| | Stim04 - Pre | −0.0060 | [−0.0094–0.0025] | 100% | Accepted | 1.00 | 24266 |
| | Stim05 - Pre | 0.0021 | [−0.0062–0.0019] | 100% | Accepted | 1.00 | 23158 |
| Beta ROPE: [0.0049–0.0040] | Stim01 - Pre | −0.0003 | [−0.0022–0.0016] | 100% | Accepted | 1.00 | 16269 |
| | Stim02 - Pre | 0.0004 | [−0.0010–0.0019] | 100% | Accepted | 1.00 | 23228 |
| | Stim03 - Pre | −0.0002 | [−0.0016–0.0013] | 100% | Accepted | 1.00 | 22834 |
| | Stim04 - Pre | 0.0006 | [−0.0008–0.0019] | 100% | Accepted | 1.00 | 23813 |
| | Stim05 - Pre | 0.0005 | [−0.0008–0.0018] | 100% | Accepted | 1.00 | 23960 |
| Gamma ROPE: [−0.0005–0.0043] | Stim01 - Pre | 0.0012 | [0.0003–0.0021] | 100% | Accepted | 1.00 | 21781 |
| | Stim02 - Pre | 0.0010 | [0–0.0019] | 100% | Accepted | 1.00 | 20823 |
| | Stim03 - Pre | 0.0011 | [0.0001–0.0020] | 100% | Accepted | 1.00 | 22414 |
| | Stim04 - Pre | 0.0020 | [0.0011–0.0029] | 100% | Accepted | 1.00 | 21750 |
| | Stim05 - Pre | 0.0014 | [0.0004–0.0023] | 100% | Accepted | 1.00 | 21925 |

distribution is 90% CI. In all frequency bands, the 90% CIs of the weighted transitivity difference ($\Delta_T$) of all auditory stimuli as compared to pre-stimuli resting state were entirely within the ROPE (Table 3).

Similar findings were noted for general efficiency *E* (Fig. 5). The distribution of general efficiency differences $\Delta_E$ for all stimuli were completely within ROPE across all frequency bands (Table 3). All the results remain robust under different priors, different models, and different strategies of handling outliers and missing data (refer to Fig. S5 to Fig. S12 in the supplementary materials).

## DISCUSSION

MEG is known to be able to capture brain activity at a high temporal resolution but at a low spatial resolution as compared to functional magnetic resonance imaging (fMRI). In the context of our experiment which used auditory form of stimuli, using fMRI would lead to technical difficulties such as scanner noise contamination while MEG can record

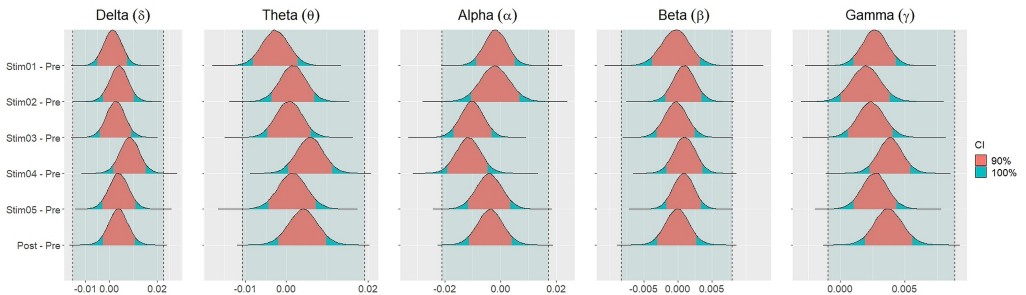

**Figure 4** Distribution of differences in transitivity *T* between each stimulus and resting state across different frequency bands.

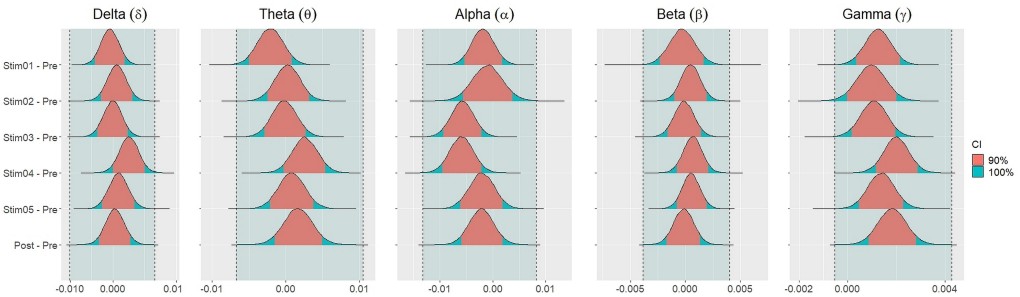

**Figure 5** Distribution of differences in global efficiency *E* between each stimulus and resting state across different frequency bands.

neural signals in silent and comfortable environment. In contrast to fMRI, it was believed that MEG and EEG might not be able to accurately detect the deep subcortical activity (*Hillebrand & Barnes, 2002*). However, an increasing number of studies have shown that MEG can detect signals from deep brain structures such as hippocampus, amygdala, thalamus and more (*Tesche, Karhu & Tissari, 1996*; *Tesche, 1996*; *Attal et al., 2007*; *Attal & Schwartz, 2013*; *Dumas et al., 2013*). Thus, coupled with depth weighted source localization method, neuronal activity at these deep subcortical structures can reliably be estimated from signals recorded at the MEG sensors. Dynamics of these brain structures are important when it comes to addressing questions pertaining to human emotional processing which are the core aspects that were probed in this study. In this study, we have included in our analysis 61 brain regions that were found to be important in emotional processing and regulations. These include superficial cortex as well as deeper subcortical structures.

In this study, we used network measures to characterize the topological features of brain networks of each subject under different auditory stimuli. Network measures quantitatively describe several aspects of functional segregation and integration, characterize the morphology of local anatomical circuitry, compute the significance of individual brain regions, and assess the resiliency of a network to insults (*Rubinov & Sporns, 2010*). Functional segregation refers to specialization of cognitive processing by densely interconnected groups of neuronal ensembles (*Sporns, 2013*). Several measures of

functional segregation have been proposed, such as, clustering coefficient and its variant transitivity. In contrast to mean clustering coefficient which is normalized individually for each node and hence be disproportionately influenced by nodes with a low degree, weighted transitivity is normalized collectively thus free from such problem (*Rubinov & Sporns, 2010*).

Functional integration on the other hand describes the capacity of the brain to combine information from distributed clusters of specialized neuronal ensembles (*Sporns, 2013*). The most widely used measure of functional integration is characteristic path length of the network. Another related measure is global efficiency which is the average inverse shortest path length between all pairs of nodes in the network. Contrary to characteristic path length, global efficiency can be computed for disconnected networks. Some authors argued that global efficiency is a superior measure of integration (*Achard & Bullmore, 2007*).

We used brain regions as mentioned in the brain region of interest section to define the vertices of the network. For the edges, we used the values of connectivity strength between the brain regions as calculated using orthogonalized AEC. Of the many connectivity measures developed and introduced in the literature, we opt for orthogonalized AEC since it is less sensitive to temporal jitter in comparison to coherence especially in higher frequency bands such as $\gamma$ band (*Bruns et al., 2000a*). Thus, AEC can capture coupling among high frequency signals which might be missed by other methods. AEC is also more apt to detect long range cortical interaction (*Bruns et al., 2000a*). AEC has been demonstrated to be one of the most consistent methods for estimating stationary connectivity in resting state experimental approach which is similar to the naturalistic paradigm used in this work (*Colclough et al., 2016*). Its repeatability is one of the reasons for its use in benchmarking other newly develop connectivity measures (*Godfrey & Singh, 2021*). For these reasons, it has been used as one of the go-to method in hyperscanning study (*Zamm et al., 2021*). Hyperscanning study is a form of neuroimaging study where the brains activities of two or more participants are recorded simultaneously whilst they interact (*Hakim et al., 2023*). For example, AEC has been used to measure interbrain coupling associated with performing musical duet (*Zamm et al., 2018*). In addition, AEC has also been used to gain deeper insight on neuropsychiatric disorders such as depression (*Nugent et al., 2020*), Alzheimer's disease (*Scheijbeler et al., 2023*), and epilepsy (*Rijal et al., 2023*). It has also been suggested as the reliable connectivity measure in monitoring the treatment efficacy in Alzheimer's disease (*Briels et al., 2020*). As mentioned before, orthogonalization is applied to the neuronal signals time series to avoid spurious coupling caused by signal leakage.

Lately, there has been a paradigm shift in the approach of how statistical analysis is done. In particular, it is widely understood that, in order to further research in all fields including neurosciences, identifying the experimental manipulation that don't cause any effects is equally important as pinpointing those that do (*Keysers, Gazzola & Wagenmakers, 2020*). Increasingly, Bayesian methods are being used to analyse data within the field of neuroscience. Using Bayesian meta-analysis, *Gambarota et al. (2022)* examined the evidences for the existence of the unconscious working memory while simultaneously uncovered several experimental variables that contribute to relevant heterogeneity in previous studies. Employing Bayesian linear regression model, *Nazari &*

*Ebersbach (2019)* found absence of an effect of distributed exercise over massed exercise in terms of mathematical performance in 7th graders after two weeks however observed positive effect in a test six weeks after. Bayesian statistics also provides a more streamline approach to analyse complex models. It has been used to fit joint models of MEG, EEG data and behaviour in order to understand cognition (*Nunez et al., 2024*; *Ghaderi-Kangavari, Rad & Nunez, 2023*). It has also been used to dissect the complex interplay between interhemispheric connectivity, endogenous GABA levels, and aging on behavioural flexibility(*Heise et al., 2022*).

In conjunction, the emphasis has now centred around parameter estimation, model comparison and model expansion instead of widely criticised point-hypothesis testing (*Gelman et al., 2013*; *McElreath, 2020*; *Kruschke & Liddell, 2018*). By fitting models that have both fixed and random effects, hierarchical or multilevel models can flexibly model complex phenomena occurring on different levels. It is useful in experiments with repeated measure design, when dealing with unequal sample size in experimental arms, or when handling observations with complex dependency structures.

Since we worked within a Bayesian framework and used linear mixed model which is a form of multilevel model, we avoided the need to control the family wise error rate for multiple comparisons (*Gelman, Hill & Yajima, 2012*). As argued in the article, multilevel models lead to the shrinkage of point and interval estimates as such the comparisons have been automatically adjusted, a phenomenon termed as 'partial pooling'. In addition, choosing a weakly informative prior provides regularization and further minimize the risk of over-fitting.

In Bayesian statistical analysis, prior is the primary means of encoding informations that are relevant to the problem being analyzed (*Gelman, Simpson & Betancourt, 2017*). These include informations about the problem that are known before considering any data or observations from the conducted study. Thus, Bayesian analysis allows for the mathematical formalism of incorporating previously accumulated scientific knowledge into the analysis *via* specifying prior distributions. Importantly, prior have to be specified in the context of the likelihood which necessitates prior predictive distribution check to ensure the plausibility of the data being generated. In essence, priors must be chosen carefully on the basis that they would generate reasonable data that are compatible with the specific domains or problems in questions. This is also important for understanding the effect of the prior on inference.

In *Gelman, Simpson & Betancourt (2017)*, the authors also argued that mindlessly applying uniform or extremely broad prior distributions as the default prior in all settings are often inappropriate and may lead to spurious inferences. The primary reason being that the vague priors may not necessarily reflect contextual knowledge of the problem domain. On the other hand, weakly informative priors have been shown to work well in practice and are thus recommended. Furthermore, estimation using weakly informative priors by incorporating background knowledge have also been proposed as one of the ways to deal with small sample size issues (*Van de Schoot & Miocević, 2020*). The term weakly informative here is taken to mean that the likelihood will dominate the prior in the presence of adequate amount of data (*Stan Development Team, 2020*). The nature of how
weakly informative a prior is must not solely be determined by how tight its distribution is but also by how it will interact with the likelihood. Thus, as argued in *Gabry et al. (2019)*, tight priors might still be considered weakly informative if they generate data that is judged extreme within the context of the domain knowledge.

Here, we specified the priors on the basis that they are weakly informative and minimally influence the results while enforcing adequate regularization in order to improve efficiency and convergence of the sampling chains. We set priors based on the range of values in the log scale of each network measures given that there were 61 brain regions involved forming connectivity matrices $\mathbf{A} \in \mathbb{R}^{61 \times 61}$. For each network measures, prior predictive checks were done to make sure that the estimated values included the entire plausible parameter space with high probability around the observed values while allowing low probability of extreme values. The priors on the slopes $\beta$ were set as stated making them unbiased to any effect direction of the stimuli on the considered network topology.

Together with prior, likelihood function is an important part that forms the generative model from which posterior distribution is derived using Bayes theorem. Referring to Eq. (10), we used log-normal distribution in our likelihood function since the network measures used as the dependent variables have positive real values and are positively skewed. In addition, it has been demonstrated that the log-normal distribution characterizes many processes and systems that comprise of many interacting parts such as brain (*Buzsáki & Mizuseki, 2014*; *Roxin et al., 2011*; *Roberts, Boonstra & Breakspear, 2015*). This phenomenon has also been observed in many other biological systems (*Zhang & Popp, 1994*). However, since log-normal distribution is not robust to outliers, we need to identify and exclude the outliers.

We set the equivalence region as stated in the statistical analysis section on the basis that resting state network is stable and does not change significantly within each subjects within the time frame of the experiment (*Chen et al., 2008*; *Demuru et al., 2017*). Furthermore, since both were resting state recordings, pre and post stimuli recordings could serve as the most logical reference or control when it comes to testing for the presence or absence of effects on brain networks which are due to auditory stimuli. The difference between pre- and post- stimuli resting state are due to other effects which are not caused by the auditory stimuli of our interest. Thus, we can safely conclude the absence of any effect for any stimuli if the difference in its network measures and the reference network measures falls within this equivalence region.

Testing for equivalence is a familiar concept and widely known in many scientific fields. Of exemplary mention is the field of pharmacology where test for bioequivalence between generic drug formulation and innovator drug formulation is routinely done. The test is carried out to ensure that the safety and the efficacy of every new generic drug is at par with the established innovator drug. Guidelines on bioequivalence study including recommendations on study design, statistical analysis and many other issues are well established continuously updated by regulatory bodies in many countries (*Midha & McKay, 2009*; *Davit et al., 2013*; *Morais & Lobato, 2010*; *Kaushal et al., 2016*). We argued that these recommendations could be adopted with suitable modifications in order to probe into the question of whether there is any effect of any stimuli on the brain networks.

On this basis, we hence follow these guidelines' convention of using 90% CIs as the intervals to test for equivalence. Defining equivalence region as stated allows for testing both the absence and presence of experimental effects. It also isolates the true experimental effects from other effects. The experimental design and the statistical approach used in this work may help expand the direction of future neuroimaging studies by enabling researchers to test for both presence as well as absence of true effects of experimental manipulations.

In this study, we were interested in exploring the potential underlying neurocorrelates of musical and other rhythmic auditory stimuli intervention. A large body of research establishes the efficacy of musical interventions in many aspects of physical, cognitive, communication, social, and emotional rehabilitation (*Standley & Prickett, 1994*). Over the past decade, neuroimaging studies have substantially advanced our understanding of how music impacts emotions. Initially, researchers primarily focused on investigating the neural substrates of emotion and how they respond to musical stimuli. These studies examined brain regions crucially involved in emotional processing and have shown how music modulates the activity in these brain regions which include the amygdala, nucleus accumbens, hippocampus, and orbitofrontal cortex (*Koelsch, 2014*; *Koelsch, 2020*). Besides, lateralization hypotheses have also been proposed comprising of the right-hemispheric dominance hypothesis which suggests that all emotions are processed in the right hemisphere, and the valence lateralization hypothesis stating that positive emotions are left-hemisphere dominant, whereas negative emotions are right hemisphere dominant (*Ocklenburg et al., 2021*; *Stanković, 2021*; *Ocklenburg, Peterburs & Mundorf, 2022*).

However, recent studies point toward the existence of multiple interrelated networks that span both sides of the brain, each associated with specific components of emotion generation, challenging the notion of strict hemispheric specialization (*Pessoa, 2017*; *Palomero-Gallagher & Amunts, 2022*; *Liu et al., 2023*). *Wu et al. (2019)* demonstrated that all 5 different musical excerpts activate network of brain regions involved in multiple cognitive functions. Notably, self-selected musical excerpt and unfamiliar musical excerpt were found to elicit largest degree of connectivity between the brain regions. Similar findings were noted by *Karmonik et al. (2016)* albeit using different musical excerpts. It was also noted that similar music creates similar functional connectivity patterns in the brain which however was modulated by musical training (*Karmonik et al., 2020*; *Niranjan et al., 2019*). A meta-analysis of fMRI data from 703 healthy subjects found that music listening coactivated multiple brain networks including limbic network that are involved in emotion processing (*Chan & Han, 2022*). However, the underlying neural mechanisms for their therapeutic effects remain poorly understood.

In light of this, *Hillecke, Nickel & Bolay (2005)* has proposed a general heuristic model, encompassing five music therapy working factors intended to serve as the foundation for further empirical studies. These factors include attention modulation, emotion modulation, cognition modulation, behaviour modulation and communication modulation. Although these factors work in tandem, one factor may play a major role as the main mechanism behind the effectiveness of music therapy in a particular disease. For example, in the case of depression, emotion modulation might play a major role in the improvement of mood. Based on the proposed heuristic model, we conducted this study to investigate the emotion

modulation factor as the potential neural mechanism of musical therapy. In particular, we focused on the topology of the brain network that have been shown to be associated with emotion processing and examined how it changes in response to five different auditory stimuli. Since, to our knowledge, this has not been done in either healthy or disease population, we initiated our investigation within healthy subjects as it could serve as the references for future research.

The results of our study showed that the topology of the brain network as characterized by the chosen graph metrics under different auditory stimuli were equivalent to the topology of the initial resting state brain network. This can be seen in all frequency bands from delta to gamma ($\delta \rightarrow \gamma$). In other words, the topology of the emotion network remains unchanged under all auditory stimuli. This finding suggests that the therapeutic effects of intervention using musical and other auditory stimuli may not be modulated through changes in the topology of emotion network. The results remain robust across diverse approaches for managing outliers and missing data, various prior specifications, and different statistical models. These results are in contrast to the findings of previous study which found that music perception is associated with enhancement in small world network organizations in the brain (*Qiu et al., 2022*; *Wu et al., 2012*).

The results, however, need to be interpreted within the context of the following study limitations. First, we have investigated the effects of auditory stimuli in healthy subjects. We argue that, since healthy individuals have optimal and resilient functional brain network topology (*Achard et al., 2006*), such short single session of exposure to auditory stimuli would not cause any noticeable changes in the topology. It would be interesting to extend the study population to include subjects afflicted with depression and anxiety. Secondly, the study consisted of only one session of 3 min length of exposure for each stimulus. Further studies are needed to investigate the effects of longer length of exposure and the effects of having multiple sessions and longer follow up. Such study would closely resemble the therapy sessions that are currently been offered in practice. Thirdly, the graph metrics that were used in this study characterize the functional brain network topology on a global scale. It is possible that the effects were more localized thus were missed by such metrics. Thus, further studies are needed to address these localized changes in the network topology. Furthermore, diverse underlying neuronal processes could possibly give rise to equivalence global network topology. Similarly, the same global network topology might be able to support diverse neuronal process.

## CONCLUSION

In this study, we sought to investigate the potential neural mechanism of music therapy focusing on the topology of the brain network involved in emotion processing in terms of transitivity and global efficiency. Specifically, we examined how both topological measures change in response to five different auditory stimuli. Employing a cross-over experimental design and Bayesian statistical analysis, we have shown that the topologies of the brain network that are associated with emotional processing in healthy subjects under these auditory stimuli were equivalent to the topology of the initial resting state brain network.

This result suggests that changes in the emotion network topology as characterized by transitivity and global efficiency may not be the underlying neural mechanism of therapy using music and other rhythmic auditory stimuli.

### Funding
This work was supported by Universiti Sains Malaysia (USM) research grants: 1002/CNEURO/910114 and 1001.PPSP.812189. The funders had no role in study design, data collection and analysis, decision to publish, or preparation of the manuscript.

### Grant Disclosures
The following grant information was disclosed by the authors:
Universiti Sains Malaysia (USM) research: 1002/CNEURO/910114, 1001.PPSP.812189.

### Competing Interests
The authors declare there are no competing interests.

### Author Contributions
- Muhammad Hakimi Mohd Rashid conceived and designed the experiments, analyzed the data, prepared figures and/or tables, authored or reviewed drafts of the article, and approved the final draft.
- Nur Syairah Ab Rani conceived and designed the experiments, performed the experiments, analyzed the data, prepared figures and/or tables, authored or reviewed drafts of the article, and approved the final draft.
- Mohammed Kannan analyzed the data, authored or reviewed drafts of the article, and approved the final draft.
- Mohd Waqiyuddin Abdullah performed the experiments, analyzed the data, authored or reviewed drafts of the article, and approved the final draft.
- Muhammad Amiri Ab Ghani conceived and designed the experiments, prepared figures and/or tables, authored or reviewed drafts of the article, and approved the final draft.
- Nidal Kamel conceived and designed the experiments, authored or reviewed drafts of the article, and approved the final draft.
- Muzaimi Mustapha conceived and designed the experiments, authored or reviewed drafts of the article, and approved the final draft.

### Human Ethics
The following information was supplied relating to ethical approvals (*i.e.,* approving body and any reference numbers):

Institutional Human Research Ethics Committee of Universiti Sains Malaysia (FWA Reg No: 00007718; IRB Reg No: 00004494).

### Data Availability
The raw MEG data are available in OpenNeuro database: Nur Syairah Ab Rani and Nurfaizatul Aisyah Ab Aziz and Mohammed Farouq Reza and Muzaimi Mustapha (2022). BRAR_NQ. OpenNeuro. [Dataset] doi: 10.18112/openneuro.ds004012.v1.0.0.

## Supplemental Information

Supplemental information for this article can be found online at http://dx.doi.org/10.7717/peerj.17721#supplemental-information.

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
