# Peer review of "Emotion brain network topology in healthy subjects following passive listening to different auditory stimuli"

_PeerJ, doi:10.7717/peerj.17721_

## Round 0.1 · original submission · Major Revisions

Dear authors:

First of all, my apologies for the delay in making a decision. I have some concerns about your manuscript and at the same time questions arise that would be of interest to the scientific community.

It has been a difficult decision but I believe it will be fair. Please pay attention in detail to the questions about methodology that each of the reviewers suggest, because I believe that many of them could be of interest.

Thank you for trusting PeerJ

Dr. Manuel Jimenez

**Language Note:** PeerJ staff have identified that the English language needs to be improved. When you prepare your next revision, please either (i) have a colleague who is proficient in English and familiar with the subject matter review your manuscript, or (ii) contact a professional editing service to review your manuscript. PeerJ can provide language editing services - you can contact us at [email protected] for pricing (be sure to provide your manuscript number and title). – PeerJ Staff

·

Basic reporting

Sentence at line 50 (‘among others’) is confusing, suggest authors to correct it.

Problem statement was clearly stated at line 73-77.

This article was clearly written, making it accessible for a wide range of readers. The language is concise and effectively conveys the key points.

Experimental design

Method of this study was stated one by one and was clearly explained in arranged form.

Amplitude envelope correlation (AEC) was used to estimate bivariate connectivity. This method has been reported by several other researchers in cognitive neurosciences.

Validity of the findings

The result of values of network measures were presented clearly by the boxplot across varies frequency bands.

The sentence in line 340, ‘Increasingly however’ was confusing. Suggest to correct the grammar.

The discussion was well-explained containing the objective of this study, results, strength and limitation, previous study. However, there is lack of works that had been done by neuroscientists on the use of Bayesian method, and AEC. Suggest to improve this part.

The conclusion was lengthy. Suggest to improve it.

Reviewer 2 ·

Basic reporting

In the introduction I suggest putting more recent references, for example from 2020 to the present, only have 1 reference to 2019 (Wu et al., 2019), the other are older than 2019.

Experimental design

Good, no comment.

Validity of the findings

Good, no comment.

Reviewer 3 ·

Basic reporting

The article is clear, the literature reference i suitable even though most of the network literature is pretty old. The manuscript has a professional and suitable structure, the figures are complete with captions.
The introductions stresses the importance of the experiment in the context of validating musico-therapy. However the scope of the experiment seems to be tangential to what was stated in the introduction.

Experimental design

The primary research is in the scope of the journal and the experiment seems to be well motivated, rigorous and performed under the highest ethical and technical standards.
The Methods were well described, however the topological network study was superficial and antiquated. State of the art methods would study motifs and connectivity beyond simple node statistics as done in this work. I suggest looking at the reviews to choose a new way of validating your results:

Statistical models of complex brain networks: a maximum entropy approach
Vito Dichio and Fabrizio De Vico Fallani

Null models in network neuroscience
František Váša & Bratislav Mišić

Or maybe using something like and a stockastick block model approach to keep the choice of priors used in the study relevant.

In general, a good approach when focusing on topological network study is took into generative models and use those as basis for inference analysis. There are many choices in network science literature for Bayesian based approaches (look at works by Tiago Peixoto, Leto Peel, Caterina DeBacco or Jean-Gabriel Young).

I understand that this might be asking a lot for a revision, an alternative would be to reduce the statement of importance of the network results in the introduction and conclusion.

Validity of the findings

No comment

Reviewer 4 ·

Basic reporting

Please refer to my comments in "4. Additional comments"

Experimental design

Please refer to my comments in "4. Additional comments"

Validity of the findings

Please refer to my comments in "4. Additional comments"

Additional comments

This study aims to investigate the neural correlates of musical therapy by examining the alternations in the topology of emotion brain network. To accomplish it, MEG recordings are obtained from 30 healthy subjects before and after exposures to five auditory stimuli. The results indicate the topology of the functional brain network does not change following the auditory stimuli. While potentially interesting, this work suffers from several major problems.
1. This manuscript lacks an explicit rationale. Additionally, it would be better to list the related hypotheses in this study.
2. How to decide on sample size, is it based on statistical analysis results (e.g. G*power)? How many individuals were recruited in total before exclusion?
3. Why are these 5 different auditory stimuli in table 1 selected? What criteria are used for selecting these stimuli? What is the frequency of these music?
4. Substantial information is missing in the Methods section.
5. How to validate the signal is from the targeted brain region exactly? For example, VTA.
6. Please provide more explanations regarding amplitude envelope correlation (AEC). What is the step frequency, 1Hz?
7. Why is the frequency range under 2Hz excluded?
8. What are the results for EEG and ECG? It would be interesting to compare these two kinds of signals.
9. The current representative figure is misleading. It would be better to demonstrate the amplitude changes along the stimulation in each band.
10. It is hard to discern the connectivity differences based on the current methods.
11. Since only healthy subjects are included, the conclusion is compromised unless patients are incorporated. It is highly recommended to add more groups.
12. Would extending the treatment duration or times yield better results?
13. The section of Discussion is redundant, please make it brief.

Reviewer 5 ·

Basic reporting

The manuscript is well-written, using clear and professional English language. The introduction provides a comprehensive background, establishing the context and significance of the research well.

1- I suggest that the authors expand on the discussion of their findings in relation to the existing literature. The discussion could benefit from deeper analysis of how their findings align or contrast with previous studies, especially those not directly related to music therapy but within the broader context of emotion regulation and neural mechanisms.

Experimental design

The study's design is robust, employing a cross-over experimental design and Bayesian statistical analysis, which are both suitable.

2- I suggest that the authors add a more detailed explanation of how each stimulus specifically contributes to understanding the neural correlates of musical therapy.

3- I suggest that the authors discuss the limitations they encountered in their study and how they tackled them.

Validity of the findings

Bayesian linear mixed models for data analysis provides a good framework for understanding the data.

4- I suggest that the authors dicuss the statistical power of the study, including any potential effects of sample size on the ability to detect significant differences or changes in network topology.

5- In order to improve the validity of the findings, the authors need to conduct a sensitivity analysis to check the robustness of the findings to the assumptions of the paper.

Additional comments

6- The authors can discuss regarding the future work of the project.

---

## Round 0.2 · accepted · Accept

Dear Co-Authors:

I appreciate your patience. The manuscript is of good quality and ready to be published.

A cordial greeting and congratulations.

Dr. Manuel Jiménez

Reviewer 4 ·

Basic reporting

N/A

Experimental design

N/A

Validity of the findings

N/A

Additional comments

The authors have answered to my questions.

Reviewer 5 ·

Basic reporting

See additional comments

Experimental design

See additional comments

Validity of the findings

See additional comments

Additional comments

The authors have addressed all my comments and concerns.